# *Listeria monocytogenes* Interferes with Host Cell Mitosis through Its Virulence Factors InlC and ActA

**DOI:** 10.3390/toxins12060411

**Published:** 2020-06-20

**Authors:** Ana Catarina Costa, Jorge Pinheiro, Sandra A. Reis, Didier Cabanes, Sandra Sousa

**Affiliations:** 1Group of Molecular Microbiology, i3S-Instituto de Investigação e Inovação em Saúde, IBMC, Universidade do Porto, 4200-135 Porto, Portugal; anacostaphd@gmail.com (A.C.C.); jorge.pinheiro@ucc.ie (J.P.); almeidareis.sandra@gmail.com (S.A.R.); didier@ibmc.up.pt (D.C.); 2Group of Cell Biology of Bacterial Infections, i3S-Instituto de Investigação e Inovação em Saúde, IBMC, Universidade do Porto, 4200-135 Porto, Portugal

**Keywords:** *Listeria monocytogenes*, cellular infection, virulence factors, mitosis, cell cycle progression

## Abstract

*Listeria monocytogenes* is among the best-characterized intracellular pathogens. Its virulence factors, and the way they interfere with host cells to hijack host functions and promote the establishment and dissemination of the infection, have been the focus of multiple studies over the last 30 years. During cellular infection, *L. monocytogenes* was shown to induce host DNA damage and delay the host cell cycle to its own benefit. However, whether the cell cycle stage would interfere with the capacity of *Listeria* to infect human cultured cell lines was never assessed. We found here that *L. monocytogenes* preferentially infects cultured cells in G2/M phases. Inside G2/M cells, the bacteria lead to an increase in the overall mitosis duration by delaying the mitotic exit. We showed that *L. monocytogenes* infection causes a sustained activation of the spindle assembly checkpoint, which we correlated with the increase in the percentage of misaligned chromosomes detected in infected cells. Moreover, we demonstrated that chromosome misalignment in *Listeria*-infected cells required the function of two *Listeria* virulence factors, ActA and InlC. Our findings show the pleiotropic role of *Listeria* virulence factors and their cooperative action in successfully establishing the cellular infection.

## 1. Introduction

*Listeria monocytogenes* (*Lm*) is a major human foodborne pathogen [1] that causes a range of pathologies, from mild gastroenteritis to septicemia, fatal infections of the central nervous system and abortions [2,3]. To cause infection, *Lm* has evolved an arsenal of nearly 50 virulence factors [4] with specific functions often mimicking host proteins, to exploit basic cell biology processes and benefit bacterial infection [5,6]. The *Lm* infection cycle in cultured cell lines has been described and the contribution of *Lm* virulence factors to infection was reported at the molecular level [6]. In particular, several studies showed that different stages of *Lm* cellular infection are dependent on the functional hijacking of the host cytoskeleton [7]. To invade epithelial cells and to disseminate within cell monolayers and tissues, *Lm* exploits actin [8], keratins [9] and tubulin [10]. ActA and InlC are *Lm* virulence factors that play key roles in bacterial dissemination by hijacking cytoskeleton components and interfering with cortical tension. ActA is a transmembrane protein exposed polarly at the surface of *Lm*. It functionally mimics the host Wiscott-Aldrich syndrome protein (WASP) and directly recruits the Arp2/3 actin nucleation complex to the bacterial surface, stimulating the actin polymerization at one pole and leading to the formation of *Lm* actin-comet tails [11,12]. The local polymerization of actin at one pole of *Lm* allows its intracellular movement and dissemination to neighboring cells [13]. In addition to actin, tubulin is also recruited to the *Lm*-associated comets and contributes to the efficient cell-to-cell spread [10]. InlC is a *Lm* secreted protein [14] shown to regulate membrane protrusion formation in polarized cells [15]. Once secreted into the host cell cytoplasm, InlC interacts with the host protein Tuba, a host scaffold protein that interacts with N-WASP at intercellular junctions to stimulate actin polymerization and control the morphology and the maintenance of the apical complex [16]. The interaction of InlC with Tuba displaces N-WASP and induces the relaxation of cortical actin tension, which improves *Lm* ability to form protrusions and efficiently spread from cell-to-cell [14,15].

During cellular infection, *Lm* largely interferes with the host cell cycle progression causing the overall increase of its duration, which correlates with an accumulation of cells in S- and G2/M-phases [17]. We aimed here to assess whether *Lm* preferentially infect cells in a particular cell cycle stage and uncover the molecular basis of the specific interaction of *Lm* with cells in G2- and M-phases, previously reported during long infections [17]. Our data shows that *Lm* preferentially infects cultured cells in the G2/M-phases of the cell cycle and increases the overall mitosis duration in these cells. The increased mitosis duration relates with *Lm*-induced chromosome misalignment at the metaphase plate, which activates host control pathways that delay mitosis. We found that both ActA and InlC promote chromosome misalignment through their interaction with the host cytoskeleton. Our data suggest that by interfering with the cell cytoskeleton and modulating intracellular cortical tension, ActA and InlC perturb the host cell cycle progression.

## 2. Results

### 2.1. Lm Preferentially Infects Cells in G2/M-Phases of the Cell Cycle

To evaluate if *Lm* preferentially invades cells in specific cell cycle stages, we infected asynchronous human epithelial intestinal (Caco-2) and placental (Jeg-3) cell lines with *Lm* constitutively expressing green fluorescent protein (*Lm*-GFP). Single cell suspensions of infected cells were sorted discriminating GFP-positive from GFP-negative cell populations. Cells containing intracellular bacteria were GFP-positive (Inf GFP+), whereas the GFP-negative (Inf GFP-) correspond to the bystander cells coming from an infected flask but with undetectable levels of intracellular bacteria. Purity of Inf GFP+ samples was 92.3 ± 1.4% for Caco-2 and 73.7 ± 3.7% for Jeg-3 cells (Figure 1A). Cells from a non-infected flask (NI) were used as the control. In both cell types, GFP-positive cells showed an increased frequency in the G2/M-phases of the cell cycle, as compared to both GFP-negative and NI cells (Figure 1B,C). Together with increased percentages of cells in G2/M-phases, Inf GFP+ cells showed a decreased frequency of G1/G0-phases whereas the percentage of cells in S-phase remained unchanged (Figure 1B,C). The frequency of cells in G2/M-phases was consistently lower in Inf GFP- than in NI samples, indicating a partial depletion of cells in G2/M-phases in Inf GFP- along with their enrichment in Inf GFP+ samples. Altogether, these results indicate that *Lm* is able to infect cells in any stage of the cell cycle and suggest its preferential targeting of G2/M-phases over S-, G1- and G0-phases of the host cell cycle.

### 2.2. Cellular Infection by Lm Increases the Duration of Host Cell Mitosis

We assessed whether *Lm* infection would interfere with progression of mitosis. As both Caco-2 and Jeg-3 cells behaved similarly, we only used Caco-2 cells. Asynchronous Caco-2 cells were infected with *Lm*-expressing GFP and followed by time-lapse microscopy (Appendix A). The cellular remodeling events taking place during mitosis were monitored in non-infected (NI) and *Lm*-infected cells (Figure 2A). We measured the duration of mitosis by quantifying the time elapsed between nuclear envelope breakdown (NEB) and anaphase onset (AO). *Lm*-infected cells (Inf) completed mitosis on average within 55.5 min (Appendix A), which corresponded to a 1.5-fold increased duration as compared to non-infected cells (NI, 39.3 min, Appendix A) (Figure 2B). These data show that intracellular *Lm* delays the progression of mitosis, which could contribute to the overall increased cell cycle duration of infected cells [17] and to the reported accumulation of cells in G2/M-phases.

The prolonged mitotic duration suggested a sustained activation of the spindle assembly checkpoint (SAC), which controls the progression from metaphase to anaphase [18]. To evaluate if SAC activation could be responsible for the increased mitosis duration observed in *Lm*-infected cells, we measured mitosis duration in NI and *Lm*-infected Caco-2 cells in the absence or presence of MPS1 inhibitor (MPS1i), which abrogates SAC function [18]. Given that the mitotic index of Caco-2 cells was very low (approx. 4% in the whole cell population, Appendix A) and that not all the cells were infected, Caco-2 cells were synchronized in the G2 to M transition with RO-3306, a CDK1 inhibitor (Figure 2C). This led to a three-fold increase in G2/M-phase cells (Appendix A). *Lm* was then allowed to invade cells and replicate intracellularly in the presence of RO-3306. After the release from cell cycle arrest, the progression to mitosis was followed by time-lapse microscopy. The increased mitosis duration found in non-synchronized infected cells (Figure 2B) was confirmed in synchronized infected cells which took longer to accomplish mitosis (120.9 ± 4.7 min) than NI cells (93.2 ± 0.5 min) (Figure 2C,D). As expected, in the presence of the MPS1 inhibitor, mitosis occurred at faster pace due to SAC override (Figure 2D) and happened at similar rates in NI and *Lm*-infected (Figure 2D). Together, these data show that intracellular *Lm* causes an increase in mitotic duration of host cells by delaying SAC resolution.

### 2.3. Lm-Infected Cells Display a Delayed Mitotic Exit

We further evaluated the progression from M- to G1-phase in NI and *Lm*-infected cells. Caco-2 cells arrested in early mitosis (prophase) by nocodazole incubation were infected with *Lm* and cell cycle profiles of NI and *Lm*-infected cells determined by FACS analysis after cell cycle arrest release for different periods of time (Figure 3A). While in asynchronous conditions approximately 10% of the cells were in G2/M-phases (Figure 3B), after nocodazole arrest this percentage raised to 60–70% (Figure 3B, NI 0 h), thus confirming the efficacy of the synchronization conditions.

The rate of infection was evaluated by FACS analysis determining the percentage of GFP+ cells, corresponding to cells with intracellular GFP-expressing *Lm*. The percentage of infected cells was rather low and remained unchanged over the time of infection (Appendix A). The DNA histograms generated by FACS analysis at 2 h post nocodazole wash out and their respective quantification (Figure 3B and Appendix A) showed that, the percentage of cells in G2/M-phases decreased two-fold in NI conditions, while in *Lm*-infected samples the decrease was reduced to 1.6-fold. In parallel, the percentages of cells in G1/G0-phases were increased by eight-fold in NI samples and by only 4.5-fold in *Lm*-infected conditions. These observations indicate that *Lm*-infected cells are hindered in the exit from G2/M-phases. To further corroborate these results, we evaluated the levels of cyclin B, whose degradation during mitosis is a key event to control mitotic exit [18]. Western Blot (WB) analysis of the different samples showed that the levels of cyclin B slightly decreased over the 2 h of release both in NI and *Lm*-infected cells (Figure 3C,D). However, *Lm*-infected cells displayed higher levels of cyclin B than the NI synchronized control cells. These data support our observation showing that *Lm*-infected cells take longer time periods to accomplish mitosis and suggest that *Lm* interferes with molecular events that sustain SAC activation and limit cyclin B degradation.

### 2.4. Lm Infection Increases the Percentage of Misaligned Chromosomes

To assess the cause of the *Lm*-induced delay in mitosis, we examined the structure of the mitotic spindle and the positioning of the chromosomes in *Lm*-infected cells by immunofluorescence microscopy. Non-infected cells were used as the control. The structure of the mitotic spindle appeared similar in NI and *Lm*-infected cells (Figure 4A). However, *Lm*-infected samples showed a 2-fold increase in the percentage of cells with chromosome misalignment (Figure 4B). While the percentage of misalignment was 18.2 ± 0.9% in NI conditions, it increased to 36.8 ± 2.7% in *Lm*-infected samples. Similar results were obtained in the placental cell line, BeWo, in which chromosome misalignment increased from 15.5 ± 2.3% in NI cells to 29.5 ± 5.9% under *Lm* infection (Appendix A). These data indicate that, during infection *Lm* perturbs the positioning of the chromosomes at the metaphase plate, increasing the number of misaligned chromosomes which would sustain SAC activation to prevent premature chromosomal migration to the poles and thus increase the mitosis duration.

### 2.5. Lm Impairs Chromosome Alignement through ActA and InlC

Chromosome alignment at the metaphase plate is a critical step in mitosis [19]. Both chromosome alignment and separation are governed by tension and physical forces developed at the level of the mitotic apparatus, their impairment often causes the appearance of non-aligned lagging chromosomes, a sustained SAC activation and ultimately mitotic delay [20]. We thus hypothesized that, during intracytoplasmic life, *Lm* may express bacterial proteins that interfere with host cytoskeleton and/or cytoplasmic tension thus promoting chromosome misalignment, impairing the mitotic exit and increasing the overall duration of mitosis. Two *Lm* virulence factors are well described to interfere with host cytoskeleton: ActA allows the formation of actin- and tubulin-comet tails that propel *Lm* across the host cytoplasm [10,21], and InlC decreases host cortical tension facilitating *Lm* cell-to-cell dissemination [15]. We assessed whether intracellular *Lm* could increase chromosome misalignment through ActA and/or InlC. Caco-2 cells were left non-infected (NI) or infected with wild-type *Lm* (WT) or isogenic deletion mutants for *actA* (∆*actA*), *inlC* (∆*inlC*) or the double mutant ∆*actAinlC*. Chromosome misalignment was quantified by fluorescence microscopy. As shown in Figure 4B, the percentage of Caco-2 cells displaying misaligned chromosomes increased 2-fold in *Lm*-infected as compared to NI samples (Figure 5A). While infection with *Lm* ∆*actA* only partially reverted the increased chromosomal misalignment, in the absence of InlC, the percentage of cells showing misaligned chromosomes dropped to levels detected in NI cells (Figure 5A). Similar data were obtained for infections performed in BeWo cells (Appendix A). These data thus indicate that, during infection, *Lm* perturbs chromosomal positioning at the metaphase plate through both ActA and InlC. During mitosis, tension applied to the mitotic spindle depends on astral microtubules that connect the centrosome to the cortex of the cells. Perturbed astral microtubules induce alterations on the spindle orientation and on the distance between the two centrosomes. We assessed whether *Lm* infection could interfere with astral microtubules by determining the angle of the mitotic spindle and the distance between centromeres. A slight increase in the mitotic spindle angle was detected in *Lm*-infected cells, but this difference was not dependent on the expression of ActA and/or InlC (Appendix A). No differences were detected in the distance between the two centrosomes (Appendix A). This suggests that *Lm* does not significantly interfere with astral microtubules.

InlC attenuates cortical tension inside cells by binding the host protein Tuba [15]. This interaction specifically requires InlC F146 and K173 amino acids [15,22]. We evaluated whether the role of InlC in chromosome positioning was related to its capacity to bind Tuba. For this, we used different *Lm* ∆*inlC* strains complemented with WT InlC or InlC F146A or InlC K173A. Caco-2 cells were infected with these strains as described and chromosome misalignment was scored. Cells infected with the strain complemented with the WT InlC (+InlC) showed similar percentages of chromosome misalignment to those infected with the WT *Lm* (Figure 5A,B). However, complementation with InlC F146A or InlC K173A did not restore *Lm* capacity to interfere with chromosome positioning. These data indicate that InlC-induced chromosome misalignment requires InlC interaction with Tuba.

## 3. Discussion

Here, we reported that, in cultured cell monolayers, *Lm* preferentially infects G2/M cells and once inside such cells, it interferes with cell cycle progression, postponing mitosis exit and thus increasing the duration of mitosis. This effect is likely to be related to the increased misalignment chromosomes detected in infected cells, driven by two *Lm* virulence factors—ActA and InlC.

*Lm* infection was previously shown to interfere with the cell cycle progression of host cells [17,23]. In these studies, *Lm* cellular infection was demonstrated to induce host DNA breaks and trigger non-canonical DNA damage response, which is partially dampened by the secretion of the pore-forming toxin Listeriolysin O (LLO) [17,23]. DNA breaks induced by *Lm* were found to favor bacterial dissemination and delay the host cell cycle progression without inducing cell cycle arrest or compromising cell viability [17]. Intracellularly, *Lm* thus appears able to adjust the host cell cycle progression to its own benefit: promoting dissemination and controlling its intracellular numbers to avoid cell damage. The manipulation of the host cell cycle is a common mechanism used by several bacterial pathogens to promote infection [24]. Indeed, pathogens produce a variety of cyclomodulins bearing different activities that interfere with typical host cell activity, including cell differentiation and development and that slow down cell and tissue renewal [24]. Overall, cyclomodulins generate favorable conditions for pathogen survival within the host. Besides bacteria, virus also extensively target host cell cycle regulation mechanisms to modulate host physiology and suit their high replication needs [25].

The cell cycle stage of the host cells is highly interconnected with cell metabolic and physiologic status [26,27] and thus may by itself affect the rate of infection. Indeed, while the influenza virus was reported to preferentially infect epithelial cells in G1 phase [28], *Salmonella* and *Cryptosporidium parvum* were shown to particularly target mitotic cells [29,30]. How the cell cycle stage of the host cell would impact the rate of *Lm* infection was never addressed. Our data indicate that during the early phases of infection, *Lm* preferentially targets cells in G2/M phases, and the molecular details associated with this observation were not exploited here, however several hypotheses can be proposed. Cells preparing to divide undergo alterations that massively change their biophysical properties which might facilitate bacterial invasion and replication in the host cytoplasm. In particular, mitotic cells modify their adhesive properties with neighbor cells and extracellular matrix to round up and accommodate the assembly of the mitotic spindle, and these alterations are accompanied, among others, by extensive actin and microtubule remodeling and membrane composition alterations [31,32]. As a consequence, mitotic cells display modified membrane tension and elasticity, internal pressure and cytoplasm viscosity. It is plausible that the specific biophysical properties of the mitotic cells benefit the infection by intracellular pathogens such as *Lm*. Membrane modifications might account for preferential invasion, as was described for *Salmonella* [29], and alterations in the cytoplasm may favor bacterial multiplication, intracellular movement and cell-to-cell spread [33,34,35].

In the host cell cytoplasm, *Lm* is well known to largely interfere with cell cytoskeleton. Through ActA, *Lm* hijacks the actin polymerization machinery to polymerize its actin comet tail that allows for bacterial movement and spreading [36], which also benefits from the recruitment of tubulin to the *Lm*-associated comet [10]. By interacting with elements of the host cytoskeleton through ActA, our data suggest that, during infection, *Lm* may compromise the course of important biological processes such as mitosis. Through the secretion of InlC, *Lm* modulates cortical tension in polarized epithelial cells promoting the formation of *Lm*-containing protrusion and thus favoring bacterial cell-to-cell spread and infection dissemination [15,36]. The structure of a polarized epithelium depends on intercellular junctions and cortical actomyosin bundles. In the absence of Tuba, intercellular junctions become slack due to the decrease in cortical tension [16]. During mitosis, the cortical tension is a key factor for the proper attachment and alignment of chromosomes that precede their migration to the poles of the spindle and effective cell division [37,38]. Interestingly, Tuba was reported to be critical for epithelium establishment and for proper orientation of mitotic spindle during epithelial cyst formation [39]. During *Lm* infection, InlC interacts with Tuba and displaces N-WASP from the junctions, leading to decreased cortical actomyosin and cortical tension which relax the cellular junctions and promote formation of membrane protrusion. InlC mutants (F146A or K173A) fail to bind Tuba and thus have no effect on cortical tension impairing protrusion formation [15,36]. Our data show that the secretion of InlC in the host cell cytoplasm and the consequent modulation of cortical tension through interaction with Tuba, impair chromosome alignment in mitotic cells and delay cell division. Despite both ActA and InlC being able to compromise chromosome alignment, InlC appears to have a strong effect which is probably related to the direct targeting of the cortical tension.

An overall delay in the host cell cycle progression was already showed to favor *Lm* infections, however, whether interfering with mitosis progression specifically benefits the infection or is a side effect of the activity of ActA and InlC is unknown, both hypotheses are plausible.

In addition, it would be interesting to assess if the effects reported here in 2D culture cells translate in the 3D complexity and organization of an epithelial tissue, such the ones that *Lm* targets during in vivo infection. The long-term effects of prolonged mitosis on cell fate remain undetermined and the biological significance of a small defect in spindle orientation in epithelial tissue is unknown.

Our data show that even though secreted and surface virulence determinants have major functions and play specific roles during infections, they are also pleiotropic factors subverting unsuspected cell biology processes. Overall, such multitasked bacterial factors work cooperatively to control the host promoting infection.

## 4. Materials and Methods

### 4.1. Bacterial Strains, Cell Lines and Growth Conditions

*Listeria monocytogenes* (*Lm*) EGDe was used to construct all the mutants used here. *Lm*-GFP (constitutively expressing GFP) [17], *Lm*∆*actA* [17], EGDe∆*inlC*, EGDe∆*actAinlC*, EGDe∆*inlC*+*inlC*, LmΔ*inlC*+i*nlC* F145A, LmΔ*inlC*+*inlC* K172A (from this work) were grown in Brain Heart Infusion (BHI, Difco Laboratories, Detroit, MI, USA) broth at 37 °C under aerobic conditions with shaking (200 rpm). Human colorectal adenocarcinoma cell line Caco-2 (ATCC HTB-37) was propagated in Eagle’s Minimum Essential Medium (EMEM) supplemented with 20% (v/v) fetal bovine serum (FBS, Biowest, France), 1 mM sodium pyruvate and 0.1 mM non-essential amino acids. Jeg-3 (ATCC HTB-36) cells from human placental origin were sub-cultured in EMEM supplemented with 10% (v/v) FBS. Human choriocarcinoma BeWo cells (ATCC CCL-98) were grown in DMEM/HamF12 in a proportion of 50/50 supplemented with 10% FBS and 2 mM glutamine. Cells were kept at 37 °C with 5% CO_2_ in a humidified atmosphere achieved by humidity pans filled with sterile distilled water, thus producing 90–95% humidity through passive evaporation inside the incubator (Binder, Neckarsulm, Germany). All the cell culture media and supplements were purchased from Lonza (Basel, Switzerland). Synchronization in G2 to M transition was achieved by incubating cells with 10 mM RO-3306 (Sigma-Aldrich, St Louis, MO, USA) for 20 h. Nocodazole (0.1 µg/mL, Sigma-Aldrich, St Louis, MO, USA) was added to the culture medium for 20 h to synchronize cells in M phase.

### 4.2. Construction of Deletion Mutants for InlC and Complemented Strains

*Lm* ∆*inlC* deletion mutants were constructed in the *Lm* EGDe strain through a double homologous recombination process mediated by the plasmid pMAD [40] as described by [41]. To create the double deletion mutant EGDe∆*actAinlC*, we used our previously constructed EGDe∆*act* deletion mutant and further deleted *inlC* [17]. DNA fragments corresponding to the upstream (UP) and downstream (DW) flanking regions of the gene *inlC* were amplified by PCR from *Lm* EGDe chromosomal DNA, using specific primers #1 plus #2 and #3 plus #4, respectively, listed in Table 1. Complementation was mediated by the *Lm* specific integrative plasmid pIMK [42], through the construction of pIMK (inlC) using primers #5 plus #6 (Table 1). To create the InlC amino acid substitutions (F146A and K173A), site-directed mutations were performed with primer pairs #7 plus #8 and #9 plus #10, respectively, listed in Table 1.

### 4.3. Infection Assays

Caco-2, Jeg-3 or BeWo cells were trypsinized and seeded in triplicate in fresh medium in 60-mm dishes or 6-well plates (Nunc, Thermo Fisher Scientific, Waltham, MA, USA) and propagated for 48 h. Bacteria were grown to an OD_600_ of 0.6–0.8, washed and diluted in EMEM. Cells were left non-infected (as control) or incubated with *Lm* or the referred isogenic mutants at multiplicity of infection (MOI) of 10-30. After invasion (1 h), cells were incubated in their respective propagation medium (indicated in 4.1) supplemented with 50 µg/mL gentamicin (Sigma-Aldrich, St Louis, MO, USA).

### 4.4. Flow Cytometry Procedures

Caco-2 cells were resuspended in sorting buffer (Ca/Mg free Phosphate Buffered Saline (PBS, Lonza, Basel, Switzerland), 1mM EDTA (Lonza, Basel, Switzerland), 25mM HEPES (Lonza, Basel, Switzerland) and 2% FBS) and sorted according to their mean GFP fluorescence intensity (GFP+ MFI), after exclusion of debris, cell doublets and dead cells. To generate DNA histograms, cells were resuspended in PBS, fixed in 70% ethanol and incubated (37 °C, 3 h) with 40 µg/mL PI and 10 µg/mL RNase A (both from Sigma-Aldrich, St Louis, MO, USA). At least 10,000 gated events were acquired in a FACS Canto II flow cytometer (BD Biosciences, San Jose, CA, USA). DNA histograms were obtained on a linear-scale PE-A histogram, after exclusion of debris and cell doublets. Data were analyzed using FlowJo software (version 9.5.2, TreeStar, Inc, Ashland, OR, USA.) and the Watson pragmatic model to quantify the percentage of cells in each cell cycle phase. Caco-2 cells were synchronized in M-phase by incubation with nocodazole (0.1 µg/mL) for 20 h and infected with *Lm* (MOI 20) for the last 5h of synchronization. After nocodazole wash out, the NI and *Lm*-infected cells were allowed to progress for additional 1 h, 2 h and 4 h. Cells were then processed for FACS as indicated above, and for immunoblot analysis (detailed in 4.6). Asynchronous cells were used as control.

### 4.5. Time-Lapse Microscopy

Caco-2 cells were trypsinized and seeded in triplicate in fresh medium (2 × 10^4^ cells) in Ibitreat μ-dishes (Ibidi GmbH, Grafelfing, Germany), grown for 24 h and left uninfected or infected with *Lm* at MOI 10. After invasion, cells were cultured in gentamicin-supplemented phenol red-free medium and followed-up by live-cell imaging. Cells were arrested in G2 to M transition by incubation with the CDK1 inhibitor RO-3306 (10 µM) for 20h. In the last 2.5 h of arrest, cells were infected with *Lm*-GFP (MOI 10, 1 h of invasion and 1.5 h of intracellular multiplication). After RO-3306 wash out, cells were incubated in the presence of an MPS1 inhibitor (MPS1i, 2 µM) or DMSO (vehicle). Images were acquired in an inverted epi-fluorescence Axiovert 200M microscope (Zeiss, Oberkochen, Germany) equipped with a NanoScan Piezo Z stage (Prior Scientific Instruments, Cambridge, UK). Cells were maintained at 37 °C in a 5% CO_2_ atmosphere. Shutters, filter wheels and point visiting were driven by Micro-Manager 1.4 software [43] and images captured with a CoolSNAP HQ camera (Roper Scientific, Trenton, NJ, USA). Phase contrast images were acquired every 2 min and GFP signal images every 20 min, at multiple points with a 20× (0.30 NA) objective. ImageJ software [44,45] was used to compile images, merge phase contrast with GFP signal and analyze the resulting movies.

### 4.6. Immunoblot Analysis

Caco-2 cells were directly lysed in Laemmli buffer (3% (v/v) glycerol, 5% (v/v) β-mercaptoethanol, 2% (v/v) SDS, 0.1% (w/v) bromophenol blue in 100 mM Tris-HCl, pH 6.8). Samples were boiled in a hot block at 95ºC for 5 min, resolved by SDS-PAGE (10% polyacrylamide gel) and proteins transferred onto nitrocellulose membranes (Bio-Rad Laboratories, Hercules, CA, USA). Membranes were blocked in 5% skimmed milk in TBS-T (20 mM Tris-HCl, pH 7.4, 137 mM NaCl, 0.1% Triton X-100), immunoblotted with primary antibodies recognizing cyclinB1 (Sigma-Aldrich, St Louis, MO, USA) and GAPDH (sc-32233, Santa Cruz Biotechnology, Dallas, TX, USA) and horseradish peroxidase-conjugated secondary antibodies (BI2413C and BI2407, PARIS Biotech, Compiègne, Paris, France). Chemiluminescence (Pierce ECL Western Blotting Substrate, Thermo Fischer Scientific, Waltham, MA, USA) was used for protein detection, and band intensities were quantified using Image J software.

### 4.7. Immunofluorescence

Caco-2 or BeWo cells on fibronectin-coated coverslips were fixed with 3% Paraformaldehyde (PFA, Sigma-Aldrich, St Louis, MO, USA) and quenched with 50 mM NH_4_Cl (20 min), permeabilized with 0.3% (v/v) Triton X-100 (5 min), washed in PBS and incubated with blocking solution (1% BSA and 20% FBS in PBS) for 30 min. Coverslips were incubated with anti α-tubulin or anti gamma-tubulin (T5168 and T5192, respectively, Sigma-Aldrich, St Louis, MO, USA), washed and incubated with AlexaFluor Cy3- (Jackson ImmunoResearch, West Grove, PA, USA) conjugated secondary antibodies, FITC-phalloidin (Thermo Fischer Scientific, Waltham, MA, USA) and 2 ng/mL 4,6-Diamidino-2-phenylindole dihydrochloride (DAPI, Sigma-Aldrich, St Louis, MO, USA). Coverslips were mounted with Aqua-Poly/Mount medium (18606, Polysciences, Warrington, PA, USA).

### 4.8. Fluorescence Microscopy and Image Analyses

To quantify spindle angle and interpolar distance, we used a Zeiss AxioImager Z1 microscope (Zeiss, Germany) equipped with a 63× (1.4NA) objective (Zeiss, Germany). Z-stack images were acquired and deconvoluted using Huygens software (Scientific Volume Imaging, Netherlands) to decrease blurring and noise. Gamma-tubulin staining of centrosomes was used as reference. For quantifications of misaligned chromosomes, an Olympus BX53 microscope (Olympus, Japan) coupled with a 20× (0.17NA) lens (Olympus, Japan) was used to obtain full coverslip imaging by multiple image alignment (MIA). ImageJ software was used for single, MIA and Z-stack images manipulation and analyses.

### 4.9. Statistical Analyses

Statistical analyses were performed with Prism 7 software (GraphPad software, Inc., San Diego, CA, USA). Two-tailed Student’s t-test was used for comparison of means between two samples. One-way ANOVA with Bonferroni’s multiple comparison test was used for pair-wise comparison of means from three or more unmatched groups. Differences between samples were considered statistically significant for *p* value < 0.05.

## Figures and Tables

**Figure 1 toxins-12-00411-f001:**
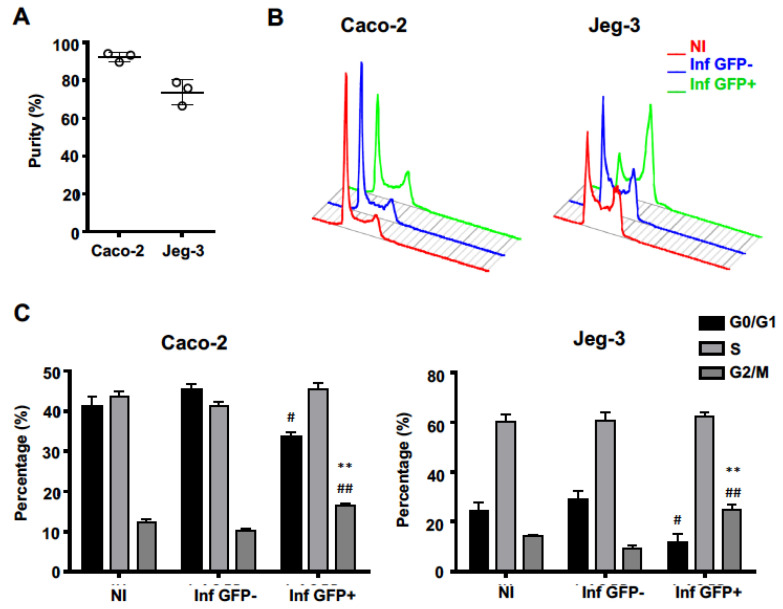
*Lm* preferentially infects cells in G2/M-phases of the host cell cycle. Caco-2 and Jeg-3 cells were infected with *Lm* expressing GFP (Multiplicity of infection, MOI 20 and 30, respectively) and sorted discriminating GFP-positive (Inf GFP+) from GFP-negative (Inf GFP-) cells. (**A**) Displays the purity of sorted GPF+ populations, from three independent experiments. (**B**) DNA histograms for different cell populations were obtained by flow cytometry (FACS) analyses and quantified (**C**) applying Watson pragmatic algorithm. (**B**) shows data from a representative experiment. In (**C**) data are means ± SEM from three independent experiments. * Indicates statistical comparisons to NI; # Indicates statistical comparisons to Inf GFP-; #: *p* < 0.05, ** and ##: *p* < 0.01 (one-way ANOVA, Bonferroni’s multiple comparison test).

**Figure 2 toxins-12-00411-f002:**
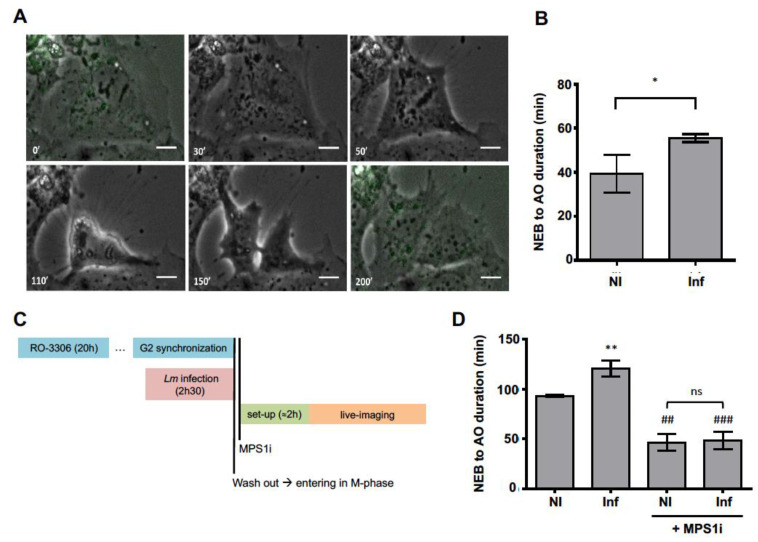
*Lm*-infected cells display an increased mitotic duration. Time lapse movies performed by acquiring bacterial GFP signal and phase contrast microscopy images (Appendix A). Movie frames illustrating mitosis are depicted in (**A**). An infected interphasic Caco-2 cell (0′) gives rise to two daughter cells (200′): first, nuclear envelope breaks down (NEB) (30′), then chromosomes fully align at the metaphase plate (50′), which is followed by anaphase onset (AO) at 110′ and, finally, cytokinesis takes place (150′). Scale bar = 15 μm. (**B**) Quantification of mitosis (NEB to AO duration) performed in asynchronous cells. Data include independent measurements obtained from more than 25 cells per independent experiment (*n* = 3). * corresponds to *p* < 0.05 (Student’s t-test). (**C**) Scheme of the experimental set-up. Caco-2 cells were arrested in G2 to M transition with CDK1 inhibitor RO-3306 (10 µM) and infected with *Lm*-GFP (MOI 10). Upon RO-3306 wash out cells were incubated with an MPS1 inhibitor (MPS1i, 2 µM) or DMSO (vehicle). (**D**) Quantification of mitotic duration (mean ± SEM) obtained from three independent experiments and 30 to 50 cells per experiment. ns stands for non-significant, ** corresponds to comparison to NI (*p* < 0.01); ## and ### correspond to comparisons without MPS1i, respectively *p* < 0.01 and *p* < 0.001 (one-way ANOVA, Bonferroni’s multiple comparison test).

**Figure 3 toxins-12-00411-f003:**
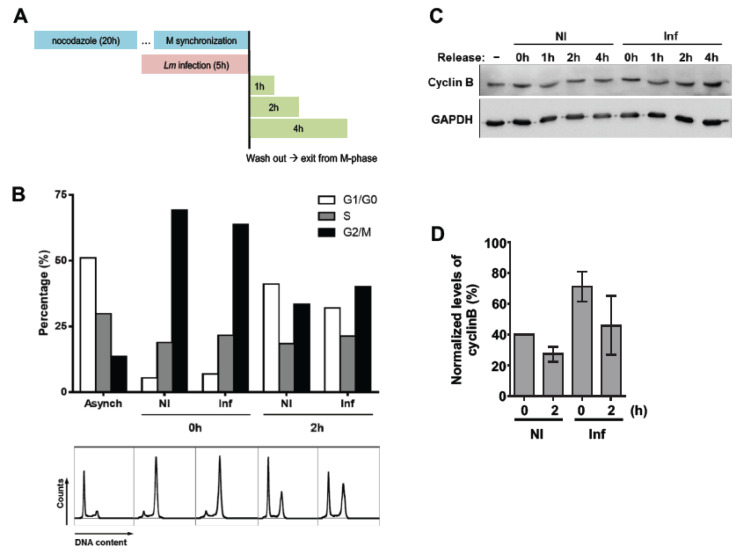
*Lm*-infected cells are delayed in mitotic exit. (**A**) Schematic representation of the experimental set-up. Caco-2 cells synchronized with nocodazole (0.1 μg/mL), were infected by *Lm* (MOI 20) and released from cell cycle arrest for different time periods. (**B**) DNA histograms (bottom panel) were generated by FACS and quantified (upper panel). Asynchronous cells were used as control. Data shown in B are representative of three independent experiments. (**C**) Immunoblot showing the levels of Cyclin B in non-infected (NI) and *Lm*-infected (Inf) Caco-2 cells before (0 h) and after (1, 2 and 4 h) nocodazole washout. GAPDH was used as a loading control to normalize protein levels. Data shown in C are representative of three independent experiments. (**D**) Quantification of immunoblots (mean ± SEM).

**Figure 4 toxins-12-00411-f004:**
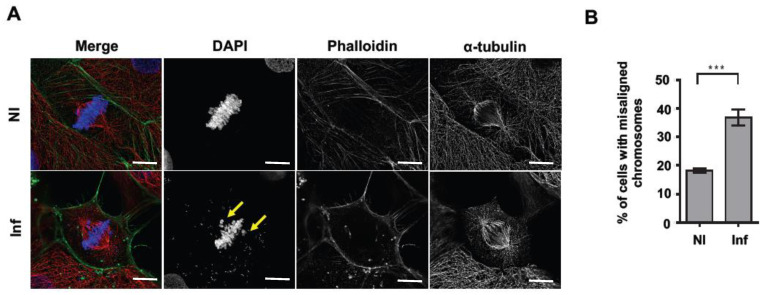
*Lm*-infected cells exhibit an increased percentage of misaligned chromosomes during mitosis. Cells were infected with *Lm* expressing GFP (green) for 7 h and stained with 4,6-Diamidino-2-phenylindole dihydrochloride (DAPI, DNA, blue), phalloidin (actin, green) and anti-α-tubulin antibody (microtubules, red). Non-infected cells (NI) were used as control. Scale bar = 20 μm. (**A**) Confocal microscope micrograph of NI and *Lm*-infected Caco-2 cells in mitosis. Arrows show misaligned chromosomes. (**B**) Quantification of cells displaying misaligned chromosomes in non-infected (NI) and *Lm*-infected (Inf) Caco-2 cells. Graph shows the mean ± SEM of four independent experiments. *** corresponds to *p* < 0.001 as obtained by Student’s t-test.

**Figure 5 toxins-12-00411-f005:**
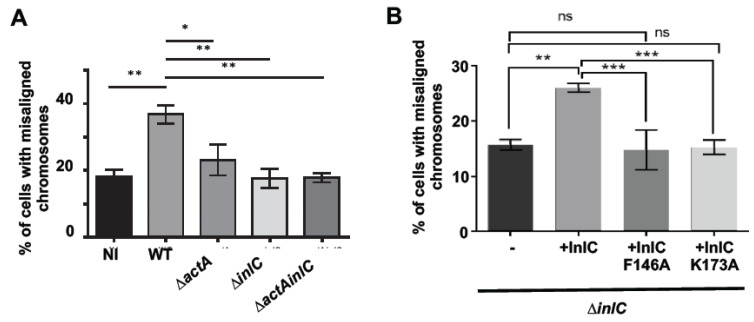
*Lm* proteins ActA and InlC perturb chromosome alignment at the metaphase plate during mitosis. Caco-2 cells were left non-infected or infected with different *Lm* strains for 7 h (MOI 30) and misaligned chromosomes were quantified. Graphs in (**A**) and (**B**) show the percentage of mitotic cells with misaligned chromosomes for each condition. Data are mean ± SEM of at least four independent experiments. ns stands for non-significant, * corresponds to *p* < 0.05, ** to *p* < 0.01 and *** to *p* < 0.001 as obtained by one-way ANOVA with Bonferroni’s multiple comparison test.

**Table 1 toxins-12-00411-t001:** Sequences of the used primers. Capital letters in primer sequences show the restriction sites.

Plasmid Generated	Primer Name	Primer Sequence (5′–3′)
pMAD (up+Dw)	# 1-Up Fwd	gcatgGTCGACggcatagaaagtagattcc
	# 2-Up Rev	agatcACGCGTcaacattctccactcctt
	# 3-Dw Fwd	cattACGCGTtaggacttgtgcacacctg
	# 4-Dw Rev	cctgAGATCTgaagtcatgttcgtcaatcg
pIMK (inlC WT)	# 5-inlC Fwd	actCCATGGcaaaaaaaaataattggttacaa
	# 6-inlC Rev	taaGGTACCctaatgatgatgatgatgatgattcttgataggttgtgtaac
pIMK (inlC F146A)	# 7-F146A Fwd	gtttatctcgcttggctttagataacaacg
	# 8-F146A Rev	cgttgttatctaaagccaagcgagataaac
pIMK (inlC K173A)	# 9-K173A Fwd	cttatctattcgtaataatgcgttaaaaagtattgtgatg
	# 10-K173A Rev	catcacaatactttttaacgcattattacgaatagataag

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
