# Peer review of "Listeria monocytogenes* Interferes with Host Cell Mitosis through Its Virulence Factors InlC and ActA"

_toxins, 2020, doi:10.3390/toxins12060411_

Round 1
Reviewer 1 Report
Dear Authors,
The manuscript “Listeria monocytogenes interferes with host cell mitosis through its virulence factors InlC and ActA” describes a series of experimental procedures involving the use of mutants for the determination of the influence of two virulence factors on the mitosis process in two human cell lines after infection.
The experiments performed sound reasonable and the results obtained are promising. However, a deep revision of the manuscript is needed for making it clear and feasible for publication.
In general, in the results section, too much details of the experimental procedures, introduction and discussion are given. This section can be dramatically shortened if these parts are moved into the dedicated sections.
The experimental part involved the use of three human cell lines for invasion assay, however, in the results section and Figure 1 only Caco-2 and Jeg-3 results are shown. Please, clarify in the text. Furthermore, it is not clear why only Caco-2 cells were used in the further experiments, please clarify in the manuscript the choices made.
In some figure legends, too many method information is repeated, sometimes giving details that are not stated in the dedicated section.
I will go through the sections of the manuscript to try to give you some guidelines to improve the presentation of the work undertaken.
L 18 Change ; with ,
Introduction
L 25 “Impressive”, poor word choice
L 27-28 Change “was” with “has been”
L 27 “Great detail”, poor word choice
L 32-33 A reference for this statement is needed at the end of the sentence
L 41 The host protein Tuba is described in the discussion section (L 299). Please move that description here
L 44 Insert “the” between “with” and “host”
L 48 “In turn”, poor word choice
L 51-56 This part belongs to the results
L 56-57 This part belongs to the discussion
Results
L 60-62 This part belongs to the introduction
L 64 What about BeWo cells? In the materials and methods it is stated that the infection assay was performed also on them
L 64-69 This part belongs to the materials and methods
L 71-73 This part belongs to the materials and methods
L 75 and 77 “Concomitantly” and “Interestingly”, poor word choice
L 78-81 This part belongs to the discussion
L 84 “MOI”, I guess refers to Multiplicity of Infection, but it has to be clarified in this first mention
L 93-94 This part belongs to the introduction
L 101-105 This part belongs to the discussion
L 108 The function of MPS1i should be described in the introduction or in the discussion.
L110-111 This part should be in the materials and methods
L 114-115 Materials and methods
L 116 The beginning of this sentence is not clear, please rephrase
L 120-121 This part should be in the discussion
L 123-138 This figure legend is too long, there is too much repetition of methods
L 125 Change “stills” with “frames”
L 131 I suggest to invert part A with part C, giving the description of the experiment first
L 141 Discussion
L 143-145 Materials and methods
L 149-152 Materials and methods
L 159-161 Discussion
L 164-166 Discussion
L 169 How the cells were synchronized should be detailed in the materials and methods
L 173 “Asynchronous cells were used as control” should be mentioned in the materials and methods, where it is actually missing
L 174 “data…Table S1”, this should be written in the text, not in the figure legend
L 181-183 Materials and methods
L 189-192 Discussion
L 203-213 This part could be either introduction or discussion
L214-217 Materials and methods
L 217 “As reported above”, please clarify
L 222-223 Discussion
L 223-226 This could be either introduction or discussion
L 231-235 Discussion
L 236-238 Materials and methods
L 242-244 Discussion
L 247-250 The description of this figure is a bit confusing, please rephrase for clarity
Discussion
L 274 “In turn” poor word choice
L 281 “can be put forward” poor word choice
L 294 “sequestering” poor word choice
Materials and Methods
In this section, information about how many technical/biological replicates and the use of controls is omitted, please add information where required.
Furthermore, check the suppliers’ information and add the Country where it is missing.
L 330 Please change “media” with “broth”
L 332 EMEM and FBS, supplier is missing
L 335 “DMEM/HamF12” needs to be specified and the supplier added
L 336 Please describe the parameters of the humidified atmosphere and how it was achieved and controlled
L 336-337 Country missing from Lonza
L 337 “Whenever stated”, too vague, please specify better
L 339 The headings from here, throughout all the following paragraphs, are numbered wrongly. Please correct.
L 340 “in a EGDe background”, please rephrase for clarity
L 341 “as described”, please change with “as described by….. [ref]
L 344 Remove “a” between “by” and “PCR”
L 345 Primers should be named as in the table (#1, #2…). Also, change “on” with “in”
L 353 Please specify with what the human cells were seeded
L 353-356 Controls and replicates not specified
L 356 Please specify which is the complete medium
L 358 Please specify which cells were used
L 358-359 In two lines, “sorting” is repeated twice, plus “sorted”. Please rephrase for clarity.
L 358 Suppliers information for buffers are missing
L 361 Please define “PBS” as it is first mentioned here (and provide supplier information)
L 363-364 Add supplier information
L 367 Specify controls and replicates in this paragraph. Also, specify what was used to seed the Caco-2 cells
L 367 Supplier information missing
L 369 “whenever stated”, please be more specific
L 371 Suppliers information missing
L 372 Specify details about the humidified atmosphere, as asked in L 336, or refer to the section 4.1
L 374-375 Supplier details and Fiji software reference missing
L 378 Please specify which cells
L 378 Laemmli buffer supplier missing
L 379 Please specify how the samples were boiled (e.g. in a water bath at 95 °C for 5 min)
L 381 Supplier information missing
L 382 “raised” poor word choice
L 383-385 Suppliers information missing
L 386 Image J and Fiji are the same software
L388 Please specify which cells
L 388 PFA, first mention, please give details and supplier
L 389 It should be NH4Cl
L 389-390 No need to specify here what PBS is, the first mention was in L 361
L 391 I suggest to put the product codes together with the supplier information (T5168 and T5192, respectively, Sigma Aldrich, Country…)
L 392-394 Suppliers information missing
L 396 Please, specify what the microscopy was used for, in both cases, at the beginning of the sentence
L 396-402 Suppliers information needed also for the microscopes, software and lenses. The reference for Fiji software should be moved to first mention in L 374
L 404 Supplier information missing
References
Please check the format and ensure that all the bacteria names are in italic.
Supplementary material
I think it is not necessary to specify in the figure number “relative to…”, as it is already stated in the manuscript.
Figure S2 The color legend is different from the graph
Figure S6 State which cell line is shown in the figure

Author Response
REFEREE 1
We want to warmly thank the reviewer for his/her comments and the carefull analysis of our manuscript. We have taken into consideration all the referee’s comments and give below a point-by-point response. Modifications in the manuscript are shown by the tracking changes tool.
In particular, we removed from the results section excessive details on experimental procedures and introduction. However, we consider that some indications are needed to explain the rationale of the proposed experiments. Concerning the fragments that the reviewer consider as being part of the discussion, in general we consider that they are the conclusion of the observations and so prefer to keep these comments in the Results section. Execessive technical details were also removed from Figure legends and added in the Material and Methods section.
Taking into account that Lm infects the intestine and the placenta during in vivo infection, in this work we used cells from both intestinal (Caco-2) and placental (Jeg-3 or Bewo) origin. Intestinal and placental cells showed to behave similarly and the majority of the experiments have been performed in Caco-2 cells, which are easy to work with and showed more consistent data. Specifically in the experiments shown in Figure 1 we used Caco-2 and Jeg-3 cells, as we found that Bewo cells were dificult to analyse by FACS. Infection of BeWo cells was performed for Supplemental Figure 4 and S5, in which we quantified chromosome misalignement.
Point-by point response to reviewer’s comments:
L 18 Change ; with ,
Changed in the new corrected version of our manuscript.
Introduction
L 25 “Impressive”, poor word choice
Changed in the new corrected version of our manuscript. “To cause infection Lm has evolved an arsenal of nearly 50 virulence factors”
L 27-28 Change “was” with “has been”
Altered in the new corrected version of our manuscript.
L 27 “Great detail”, poor word choice
Changed in the new corrected version of our manuscript. “Great detail” was deleted.
L 32-33 A reference for this statement is needed at the end of the sentence
A reference was added in the new version of our manuscript. Reference 7.de Souza Santos, M.; Orth, K. Subversion of the cytoskeleton by intracellular bacteria: lessons from Listeria, Salmonella and Vibrio. Cell Microbiol 2015, 17, 164-173, doi:10.1111/cmi.12399.
L 41 The host protein Tuba is described in the discussion section (L 299). Please move that description here
The description of Tuba was moved from the Discussion to the introduction, as suggested by the referee.
L 44 Insert “the” between “with” and “host”
Added in the new version of our manuscript.
L 48 “In turn”, poor word choice
The sentence starting with “In turn” was deleted.
L 51-56 This part belongs to the results
This part of the text was modified according to reviewer’s suggestion.
L 56-57 This part belongs to the discussion
This part of the text was modified according to reviewer’s suggestion.
Results
L 60-62 This part belongs to the introduction
Given that this idea was already mentioned in the Introduction we deleted it.
L 64 What about BeWo cells? In the materials and methods it is stated that the infection assay was performed also on them
Taking into account that Lm infects the intestine and the placenta during in vivo infection, in this work we used cells from both intestinal (Caco-2) and placental (Jeg-3 or Bewo) origin. Specifically in the experiments shown in Figure 1 we used Caco-2 and Jeg-3 cells, as we found that Bewo cells were dificult to analyse by FACS. Infection of BeWo cells was performed for Supplemental Figure 6, that’s why we include BeWo cells in infection assays in the material and methods.
L 64-69 This part belongs to the materials and methods
Modified as suggested in the new version of the manuscript.
L 71-73 This part belongs to the materials and methods
Modified as suggested in the new version of the manuscript.
L 75 and 77 “Concomitantly” and “Interestingly”, poor word choice
“Concomitantly” was replaced by “together”. “Interestingly” was deleted.
L 78-81 This part belongs to the discussion
We understand the point of the reviewer, however we consider that this is the conclusion of the observations made and think that it makes sense to keep it in the results section.
L 84 “MOI”, I guess refers to Multiplicity of Infection, but it has to be clarified in this first mention
The definition of the abbreviation MOI (Multiplicity of Infection) was added here.
L 93-94 This part belongs to the introduction
Modified as suggested in the new version of the manuscript.
L 101-105 This part belongs to the discussion
We understand the point of the reviewer, however we consider that this is the conclusion of the observations made and think that it makes sense to keep it in the results section.
L 108 The function of MPS1i should be described in the introduction or in the discussion.
We prefer to keep this information here, as we think that it facilitates the understanding of the reader. However, we simplified and reduced the information given.
L110-111 This part should be in the materials and methods
Modified in the version of the manuscript. Info on methods was partially moved to “Materials and Methods” section.
L 114-115 Materials and methods
Modified in the new version of our manuscript.
L 116 The beginning of this sentence is not clear, please rephrase
Modified in the new version of our manuscript.
L 120-121 This part should be in the discussion
We understand the point of the reviewer, however we consider that this is the conclusion of the observations made and think that it makes sense to keep it in the results section.
L 123-138 This figure legend is too long, there is too much repetition of methods
Simplified in the new version of the manuscript.
L 125 Change “stills” with “frames”
Modified in the new version of our manuscript.
L 131 I suggest to invert part A with part C, giving the description of the experiment first
The description of the experiment given in C, only relates with quantifications in D. Figure 2A and 2B concern non-synchronized cells as indicated.
L 141 Discussion
Modified in the revised version of the manuscript.
L 143-145 Materials and methods
Simplified in the revised version of the manuscript.
L 149-152 Materials and methods
Modified in the revised version.
L 159-161 Discussion
We understand the point of the reviewer, however we consider that this is the conclusion of the observations made and think that it makes sense to keep it in the results section.
L 164-166 Discussion
We understand the point of the reviewer, however we consider that this is the conclusion of the observations made and think that it makes sense to keep it in the results section.
L 169 How the cells were synchronized should be detailed in the materials and methods
Modified in the revised version.
L 173 “Asynchronous cells were used as control” should be mentioned in the materials and methods, where it is actually missing
Added, as suggested, in the “Materials and Methods” section in the new version of the manuscript. However, also kept in the figure legend where we think that is important for easy understanding.
L 174 “data…Table S1”, this should be written in the text, not in the figure legend
The modification was done as suggested by the reviewer.
L 181-183 Materials and methods
Modified in the new version of the manuscript.
L 189-192 Discussion
We understand the point of the reviewer, however we consider that this is the conclusion of the observations made and think that it makes sense to keep it in the results section.
L 203-213 This part could be either introduction or discussion
We understand the point of the reviewer. In our opinion this part of the text is required here to contextualize the experiments that are presented next. We thus prefer to keep this paragraph in the results.
L214-217 Materials and methods
Modified in the new version of the manuscript.
L 217 “As reported above”, please clarify
Modified in the new version of the manuscript.
L 222-223 Discussion
We understand the point of the reviewer, however we consider that this is the conclusion of the observations made and think that it makes sense to keep it in the results section.
L 223-226 This could be either introduction or discussion
Simplified in the new version of the manuscript.
L 231-235 Discussion
We understand the point of the reviewer, however we consider that this is the conclusion of the observations made and think that it makes sense to keep it in the results section.
L 236-238 Materials and methods
Modified in the new version of the manuscript.
L 242-244 Discussion
Modified in the revised version.
L 247-250 The description of this figure is a bit confusing, please rephrase for clarity
Modified in the revised version.
Discussion
L 274 “In turn” poor word choice
“In turn” was deleted in the new version of our manuscript.
L 281 “can be put forward” poor word choice
Replaced by “can be proposed” in the new version of the manuscript.
L 294 “sequestering” poor word choice
Replaced by “interacting with” in the new version of the manuscript.
Materials and Methods
In this section, information about how many technical/biological replicates and the use of controls is omitted, please add information where required.
We added this information here, however we also keep this info in the legend figure where it makes more sense for us.
Furthermore, check the suppliers’ information and add the Country where it is missing.
This information has been added in the new version of the manuscript (identified by track changes tool).
L 330 Please change “media” with “broth”
Modified in the new version of our manuscript.
L 332 EMEM and FBS, supplier is missing
FBS was from Biowest (France), this information has been added in the version of the manuscript. As indicated “All the cell culture media and supplements were purchased from Lonza”, we have now added the Lonza (Basel, Switzerland).
L 335 “DMEM/HamF12” needs to be specified and the supplier added
As already indicated DMEM/HamF12 were used in a proportion of 50/50. We slightly altered the text for clarity. As the other cell culture media and supplements, they wre purchased from Lonza (Basel, Switzerland).
L 336 Please describe the parameters of the humidified atmosphere and how it was achieved and controlled
Humidified atmosphere inside our Binder incubators is achieved by humidity pans filled with sterile distilled water, thus producing 90-95% humidity through passive evaporation inside the incubator (Binder, Germany). Water is changed once per week. This information was added in Materials and Methods section.
L 336-337 Country missing from Lonza
Added in the new version of the manuscript.
L 337 “Whenever stated”, too vague, please specify better
This has been specified in the new version of the manuscript.
L 339 The headings from here, throughout all the following paragraphs, are numbered wrongly. Please correct.
This has been corrected in the new version of the manuscript.
L 340 “in a EGDe background”, please rephrase for clarity
Changed in the new version of the manuscript (“in Lm EGDe strain”).
L 341 “as described”, please change with “as described by….. [ref]
Changed in the new version of the manuscript.
L 344 Remove “a” between “by” and “PCR”
Removed in the new version of the manuscript.
L 345 Primers should be named as in the table (#1, #2…). Also, change “on” with “in”
Modified in the new version of the manuscript.
L 353 Please specify with what the human cells were seeded
Specified in the new version.
L 353-356 Controls and replicates not specified
Added in the new version of the manuscript.
L 356 Please specify which is the complete medium
The complete medium corresponds to the propagation medium (described in section 4.1) for each cell type. This was specified in the new version of the manuscript.
L 358 Please specify which cells were used
Caco-2 cells, specified in the new version of the manuscript.
L 358-359 In two lines, “sorting” is repeated twice, plus “sorted”. Please rephrase for clarity.
Modified in the new version.
L 358 Suppliers information for buffers are missing
Added in the new version of the manuscript.
L 361 Please define “PBS” as it is first mentioned here (and provide supplier information)
Added in the new version of the manuscript.
L 363-364 Add supplier information
Added in the new version of the manuscript.
L 367 Specify controls and replicates in this paragraph. Also, specify what was used to seed the Caco-2 cells
Specified in the new version of the manuscript.
L 367 Supplier information missing
Added in the new version of the manuscript.
L 369 “whenever stated”, please be more specific
Modified in the new version of the manuscript.
L 371 Suppliers information missing
Added in the new version of the manuscript.
L 372 Specify details about the humidified atmosphere, as asked in L 336, or refer to the section 4.1
Specified in the new version of the manuscript.
L 374-375 Supplier details and Fiji software reference missing
Added in the new version of the manuscript.
L 378 Please specify which cells
Specified in the new version of the manuscript.
L 378 Laemmli buffer supplier missing
Laemmli buffer was homemade following the recipe indicated [3 % (v/v) glycerol, 5 % (v/v) -mercaptoethanol, 2 % (v/v) SDS, 0.1 % (w/v) bromophenol blue in 100 mM Tris-HCl, pH 6.8].
L 379 Please specify how the samples were boiled (e.g. in a water bath at 95 °C for 5 min)
Specified in the new version of the manuscript.
L 381 Supplier information missing
Added in the new version of the manuscript.
L 382 “raised” poor word choice
Modified in the new version of the manscript. “Raised” was replaced by “Recognizing”
L 383-385 Suppliers information missing
Added in the new version of the manuscript.
L 386 Image J and Fiji are the same software
Corrected in the new version of the manuscript.
L388 Please specify which cells
Specified in the new version of the manuscript.
L 388 PFA, first mention, please give details and supplier
Added in the new version of the manuscript.
L 389 It should be NH4Cl
Corrected in the new version of the manuscript.
L 389-390 No need to specify here what PBS is, the first mention was in L 361
Corrected in the version of the manuscript.
L 391 I suggest to put the product codes together with the supplier information (T5168 and T5192, respectively, Sigma Aldrich, Country…)
Modified in the new version of the manuscript.
L 392-394 Suppliers information missing
Added in the new version of the manuscript.
L 396 Please, specify what the microscopy was used for, in both cases, at the beginning of the sentence
Modified in the new version of the manuscript.
L 396-402 Suppliers information needed also for the microscopes, software and lenses. The reference for Fiji software should be moved to first mention in L 374
Added and modified as suggested.
L 404 Supplier information missing
Added in the new version of the manuscript.
References
Please check the format and ensure that all the bacteria names are in italic.
References were checked and the names of bacteria are now in italic.
Supplementary material
I think it is not necessary to specify in the figure number “relative to…”, as it is already stated in the manuscript.
Altered in the new version of the manuscript.
Figure S2 The color legend is different from the graph
Corrected in the new version of the manuscript.
Figure S6 State which cell line is shown in the figure
Cell line indicated in the new version of the manuscript.

Reviewer 2 Report
I have carefully reviewed manuscript ID toxins-820801 titled "Listeria monocytogenes interferes with host cell mitosis through its virulence factors InlC and ActA". The findings in this study provide compelling evidence of the pleiotropic role InlC and ActA play in the infection process and their influence on the host cell cycle. Carrying on from previous work (e.g. Leitao et al., 2014; Costa et al., 2019) this study is an excellent example of incremental research aimed at furthering our understanding of the way intracellular pathogens interact with host cell machinery. The methods used in this research were appropriate and the experimental design logical. Interpretation of the data appeared to be sound and well supported. Overall the manuscript is very well written and represents a strong contribution to the scientific literature. Any issues I have regarding the research and its presentation are minor in nature and are listed in my comments below.
Comments/queries
Line 6: Replace "along" with "over"
Line 8: Should be "to its own benefit."
Line 9: Change to "the capacity of Listeria..."
Line 11: Should be "exits"
Line 16: Should be "establishing the"
Line 20: Period missing at sentence end
Line 24: Delete "ranging" and place comma after "pathologies" as it is redundant since you already said it causes "a range of pathologies" on line 23..
Line 25: Should be "...has evolved..."
Lines 51-57: These comments are more in tune with the Results/ Discussion sections. Your Intro should state what the objectives of the study are but not delve into interpreting the findings.
Line 70: Generally comment - Any thoughts as to why the purity of infected Jeg-3 cells was so low relative to the Caco-2 assays?
Line 80: Reword "any stage of the cell cycle"
Line 132: Change to "...with CDK1 inhibitor RO-3306 (10 uM, 20 h)."
Line 135: How many cells on average were observed for each independent experiment?
Line 143: Delete "and" - i.e. "...(prophase), thus enriching the...")
Lines 163-164: Does this mean increased expression of cyclin B itself or is this due to impaired regulatory factors?
Line 182: Should be "the control"
Lines 219-220: Interesting...Can you speculate as to the reason for this partial effect for delta actA versus the full effect for the deletion in inlC?
Line 330: Rate of shaking?
Line 336: Subscript needed for CO2
Line 345: Change on Table to in Table
Line 355: Define acronym "multiplicity of infection (MOI)"
Line 381: Should be "membranes"
Line 406: Change "three of more " to "three or more"
Supplemental Data: Figure S2 - Your legend appears to be off as the bars on the graph are white, gray and black but legend shows two shades of gray and black labels
Author Response
REFEREE 2
We are grateful to the reviewer for his/her comments and the carefull analysis of our manuscript. We have taken into consideration all the referee’s comments and give below a point-by-point response. Modifications in the manuscript are shown by the tracking changes tool.
Point-by point response to reviewer’s comments:
Line 6: Replace "along" with "over"
Replaced in the new version of the manuscript.
Line 8: Should be "to its own benefit."
Corrected in the new version of the manuscript.
Line 9: Change to "the capacity of Listeria..."
Modified in the new version of the manuscript.
Line 11: Should be "exits"
There was a typo on line 11 that is corrected in the new version of the manuscript. “mitotic exist” was replaced by “mitotic exit”.
Line 16: Should be "establishing the"
Corrected in the new version of the manuscript.
Line 20: Period missing at sentence end
Period added in the new version of the manuscript.
Line 24: Delete "ranging" and place comma after "pathologies" as it is redundant since you already said it causes "a range of pathologies" on line 23..
Corrected in the new version of the manuscript.
Line 25: Should be "...has evolved..."
Corrected in the new version of the manuscript.
Lines 51-57: These comments are more in tune with the Results/ Discussion sections. Your Intro should state what the objectives of the study are but not delve into interpreting the findings.
This part of the text was modified in our new version of the manuscript, accordingly to reviewer’s comment. Some info was moved to Results and Discussion sections.
Line 70: Generally comment - Any thoughts as to why the purity of infected Jeg-3 cells was so low relative to the Caco-2 assays?
We believe that the differences in purity, upon cell sorting, of infected Jeg-3 and Caco-2 cells is intrinsic to the properties of the cells and the way they behave upon processing for FACS analysis. We have repeated those experiments exhaustively and Jeg-3 never reach the levels of purity of Caco-2 cells.
Line 80: Reword "any stage of the cell cycle"
Corrected in the new version of our manuscript.
Line 132: Change to "...with CDK1 inhibitor RO-3306 (10 uM, 20 h)."
Modified in the new version of the manuscript.
Line 135: How many cells on average were observed for each independent experiment?
More than 25 cells were analyzed per condition and per independent experiment. On average 30 to 50 cells were analyzed in 3 independent experiments. This is specified in the new version of the manuscript.
Line 143: Delete "and" - i.e. "...(prophase), thus enriching the...")
Modified in the new version of the manuscript.
Lines 163-164: Does this mean increased expression of cyclin B itself or is this due to impaired regulatory factors?
We believe that this is related to impaired regulation. To be degraded cyclin B needs to be ubiquitinated and displaced from its interaction with CDK1 (thus decreasing CDK1 activity). Cyclin B degradation allows the cell cycle to progress, in infected cells the “presence” of the bacteria delays the progression and so the levels of cyclin B are higher. However, we do not have experimental evidences for that.
Line 182: Should be "the control"
Corrected in the new version of the manuscript.
Lines 219-220: Interesting...Can you speculate as to the reason for this partial effect for delta actA versus the full effect for the deletion in inlC?
Here, the cortical tension is presumably the major factor affected during Lm infection. Besides the effect of ActA perturbing the host cytoskeleton, the direct targeting of the cortical tension by InlC probably plays the major role. The connection of the mitotic spindle to the cortex is of critical importance for mitosis, by directly interfering with this link InlC by itself is able produce almost the phenotype. The cell is able to partially cope with the mild cytoskeleton alterations produced by ActA in the cytoplasm.
Line 330: Rate of shaking?
Added in the new version of the manuscript (200 rpm).
Line 336: Subscript needed for CO2
Corrected in the new version of the manuscript.
Line 345: Change on Table to in Table
Modified in the new version of the manuscript.
Line 355: Define acronym "multiplicity of infection (MOI)"
Defined in the new version of the manuscript.
Line 381: Should be "membranes"
Corrected in the new version of the manuscript.
Line 406: Change "three of more " to "three or more"
Modified in the new version of the manuscript.
Supplemental Data: Figure S2 - Your legend appears to be off as the bars on the graph are white, gray and black but legend shows two shades of gray and black labels
This has been corrected in the new version of the manuscript.

Round 2
Reviewer 1 Report
Dear Authors,
After the first reviewing process, the presentation of the work undertaken appears much clearer and fluid. The use of human cell lines has been explained properly and the controls are now clear in the materials and methods section. It is my opinion that the results section is still including too much discussion and introduction, but I will leave any decision to the Editor about it.
Here following, just a few minor points to correct:
L 71 Change “is” with “was”
L 92 Change “complete” with “completed”
L 93 Change “corresponds” with “corresponded”
L 150 Change “decrease” with “decreased”
L 151 Change “display” with “displayed”
L 202 Change “reverts” with “reverted”
L 340 + 357 “In some experiments” is a bit too vague, please be more specific

Author Response
All the suggestions were included in the new version of the manuscript. Corrections are highlighted with "track changes" tool.
L 71 Change “is” with “was”
Corrected in the new version.
L 92 Change “complete” with “completed”
Corrected in the new version.
L 93 Change “corresponds” with “corresponded”
Corrected in the new version.
L 150 Change “decrease” with “decreased”
Corrected in the new version.
L 151 Change “display” with “displayed”
Corrected in the new version.
L 202 Change “reverts” with “reverted”
Corrected in the new version.
L 340 + 357 “In some experiments” is a bit too vague, please be more specific
Specified in the new version.